# Towards Sampling Data Structures for Tensor Products in Turnstile Streams

**Zhao Song**[*]           **Shenghao Xie**[†]           **Samson Zhou**[‡]

## Abstract

This paper studies the computational challenges of large-scale attention-based models in artificial intelligence by introducing innovative sampling methods in the streaming setting. Inspired by the classical definition of the $\ell_2$ sampler and the recent progress of the attention scheme in Large Language Models (LLMs), we propose the definition of the attention sampler. These attention samplers select the important coordinates in attention computation efficiently, bypassing the quadratic computational burden of computing the entire attention matrix. We demonstrate the effectiveness of the attention sampler from a theoretical perspective, including space and update time. Additionally, our framework exhibits scalability and broad applicability across various model architectures and domains.

## 1 Introduction

In recent years, the field of artificial intelligence has witnessed a significant paradigm shift with the advent of attention-based models, particularly in the domains of natural language processing and computer vision (Vaswani et al., 2017; Devlin et al., 2019; Liu et al., 2019; Yang et al., 2019; Brown et al., 2020; Zhang et al., 2022; Chowdhery et al., 2023; Touvron et al., 2023a;b; Inc., 2023; Manyika & Hsiao, 2023). At the heart of these models lies the attention mechanism (Vaswani et al., 2017), which is a powerful tool for enhancing the performance of deep learning networks. It enables models to focus on relevant parts of the input data, thereby facilitating context-aware processing.

However, as these models scale in size and complexity (Zeng et al., 2024; Reid et al., 2024; Zhang et al., 2024; Dubey et al., 2024; Abdin et al., 2024), the computational demands of the attention mechanism increase exponentially, posing significant challenges in terms of efficiency and scalability (Fu, 2024). In particular, traditional attention mechanisms used in Transformer models (Vaswani et al., 2017) require computing attention weights across all elements of the input sequence, leading to a *quadratic* increase in computational complexity with respect to the sequence length (Alman & Song, 2023; Kacham et al., 2023; Han et al., 2024; Zandieh et al., 2023; Alman & Song, 2024a;b; 2025a). This computational burden becomes particularly pronounced in large-scale applications. It hinders the usage of attention-based models in resource-constrained settings and limits their real-time processing capabilities. To deal with this problem, the core question we ask in this paper is:

*Instead of computing all entries, can we recover the most important ones in efficient space and time?*

**Attention samplers.** We adopt the classical idea of sampling a dataset, which selects important items to represent the entire dataset. Sampling is a central and effective technique for analyzing large-scale datasets, which has broad application in the field of big data (Vitter, 1985; Gemulla et al., 2008; Cohen et al., 2011; 2014), including network traffic monitoring (Mai et al., 2006; Huang et al., 2007; Thottan et al., 2010), database management (Haas & Swami, 1992; Haas, 2016; Cohen & Geri, 2019), and data summarization (Frieze et al., 2004; Aggarwal et al., 2009; Mahabadi et al., 2019; Indyk et al., 2020; Mahabadi et al., 2020). A well-known example is the $\ell_2$ sampler that is first asked by Cormode et al. (2005) and studied by Monemizadeh & Woodruff (2010): given a vector $x \in \mathbb{R}^n$, we sample an index $i \in [n]$ with probability $\frac{x_i^2}{\|x\|_2^2}$. Inspired by the $\ell_2$ sampler, we propose the following

---

[*]University of California, Berkeley. E-mail: `magic.linuxkde@gmail.com`.

[†]Texas A&M University. E-mail: `xsh1302@gmail.com`.

[‡]Texas A&M University. E-mail: `samsonzhou@gmail.com`.

attention sampler, which seeks to sample the most important coordinates in attention computation, reducing computational overhead and computer storage.

**Definition 1.1** (Attention sampler). Given matrix $A \in \mathbb{R}^{n^2 \times d^2}$, vector $x \in \mathbb{R}^{d^2}$, and a distribution function $g$, the attention sampler samples index $i \in [n]$ with probability $p_i = \frac{g((Ax)_i)}{\sum_{j=1}^n g((Ax)_j)}$.

To understand the matrix-vector multiplication $Ax$ in our attention sampler, recall that given input matrices $A_1, A_2 \in \mathbb{R}^{n \times d}$, the linear (cross-)attention matrix is defined as $QK^\top := A_1 W_Q W_K^\top A_2^\top$, where $W_Q, W_K \in \mathbb{R}^{d \times d}$ are learned weighted projections. For linear self-attention, $A_1$ and $A_2$ are identical. Let $X = W_Q W_K^\top$ be the fused key and query projection matrix, let $A = A_1 \otimes A_2 \in \mathbb{R}^{n^2 \times d^2}$, and let $x = \text{vec}(X) \in \mathbb{R}^{d^2}$, we simplify the expression by a well-known tensor product construction (Alman & Song, 2024a): $\text{vec}(QK^\top) = \text{vec}(A_1 X A_2^\top) = Ax$. Here, vec denotes the vector representation of a matrix by concatenating the rows. Therefore, our attention sampler detects the dominant entry in the linear attention matrix with respect to the distribution function $g$.

A crucial practical relevance of our attention sampler is the *sparse attention mechanisms*. The attention matrix has been shown to be naturally sparse empirically and theoretically (see e.g. (Child et al., 2019; Deng et al., 2024b)). Based on this observation, researchers seek to reduce computation by focusing on heavy coordinates (Child et al., 2019; Kitaev et al., 2020; Wang et al., 2020; Alman & Song, 2023; Brand et al., 2024; Deng et al., 2023c; Lai et al., 2025; Xiao et al., 2025; Zhang et al., 2025). In particular, they construct a *sparse mask* that selects the importance entries in the attention scheme while zeroes out the others. Since the attention sampler recovers the heavy coordinates in the sparse attention matrix, they can be applied to enhance the construction of the sparse mask.

**Attention samplers in turnstile streams.** Suppose that we can store and process the entire attention matrix, then the sampling problem is trivial since we can compute the sampling probabilities explicitly. However, as mentioned earlier, we have strict memory and process time constraints in real-world applications, which inspires us to study the attention sampler in the streaming model. In this model, an underlying dataset is defined implicitly by a data stream, in which each element of the dataset arrives sequentially. The core advantage of streaming algorithms is their ability to provide accurate estimates without storing the entire input stream, enabling real-time monitoring of large-scale datasets. Therefore, this paradigm is particularly well-suited to tackle the computational bottleneck for modern attention mechanisms. For instance, recent theoretical work (Han et al., 2025; Haris & Onak, 2025) studied attention approximation algorithms in the streaming model inspired by auto-regressive Transformers. These results closely align with the practical requirements of research in streaming Large Language Models (Xiao et al., 2024), a field that has seen a surge of interest in LLM research (Strati et al., 2024; Yao et al., 2024; Shikhar et al., 2025; Xiao et al., 2025), where the motivation comes from long (or infinite) sequence generation, e.g., a chat bot having a day-long conversation.

We study the attention sampler in the streaming model, where each entry in the input matrix $A$, the weight vector $x$, or both $A$ and $x$ is updated by a data stream, and the goal is to report a valid sample *at all times* using efficient space and update time. In particular, we consider the turnstile data streams where the entries of $A$ or $x$ can either be increased or decreased upon each streaming update. Turnstile streams naturally capture settings where data must be updated, not only appended. A key application is machine unlearning, where we need to remove or negate the contribution of specific data points without retraining the entire model. In the turnstile model, the attention sampler need to incorporate both positive updates (new tokens or training examples) and negative updates (removing or editing previous tokens, retracting user-provided data, or forgetting specific training segments).

## 1.1 OUR RESULTS

Inspired by softmax attention, where scores are defined by the exponential function $g(z) = \exp(z)$, we consider the exponential sampler that reports an index with probability $\frac{\exp((Ax)_i)}{\sum_{j=1}^n \exp((Ax)_j)}$. We show that there is a quadratic lower bound on the space complexity of exponential samplers.

**Theorem 1.2.** *[Informal statement of Theorem 4.4] Given a matrix $A \in \mathbb{R}^{n^2 \times d^2}$ and a vector $x \in \mathbb{R}^{d^2}$, where either $A$ or $x$, or both are changed by a sequence of updates. Then any algorithm that samples an index $i \in [n^2]$ with probability proportional to $p_i = \frac{\exp((Ax)_i)}{\sum_{j=1}^n \exp((Ax)_j)}$ must use $\Omega(n^2)$*

*bits of space, even if the sampling probabilities are allowed to be distorted by as large as $n^C$ for some constant $C$ and even if $\|Ax\|_\infty = O(\log n)$.*

This result demonstrates that the quadratic space complexity is unavoidable for exponential samplers, even when allowing a polynomial distortion $n^C$ to the sampling probabilities (i.e., $p_i$ suffices to be in $\left[\frac{n^{-C} \cdot \exp((Ax)_i)}{\sum_{j=1}^n \exp((Ax)_j)}, \frac{n^C \cdot \exp((Ax)_i)}{\sum_{j=1}^n \exp((Ax)_j)}\right]$). Moreover, the hard instance is established within insertion-only streams, where updates to $A$ and $x$ are limited to increments, and naturally generalizes to turnstile streams. Our findings are consistent with offline time complexity lower bounds (Alman & Song, 2023), which show that entry-wise approximations to the softmax attention matrix are unattainable in sub-quadratic time when entries exceed $O(\log n)$. In fact, our results further relax this assumption, showing that quadratic space is required for exponential samplers even if the entries are bounded.

These hardness results highlight the inherent difficulty in computing softmax attention scores. Conversely, recent work has shown both theoretically and empirically that *polynomial* attention can effectively simulate softmax while admitting more favorable algorithmic structure (Kacham et al., 2023; Koohpayegani & Pirsiavash, 2024; Saratchandran et al., 2024; Aliakbarpour et al., 2025). In particular, the PolySketchFormer (Kacham et al., 2023) demonstrates that polynomial attention achieves model quality comparable to softmax attentions with efficient low-dimensional approximations. Furthermore, polynomial attention schemes perform competitively in vision and Natural Language Processing (NLP) tasks, including the linear attention in Koohpayegani & Pirsiavash (2024) and the polynomial attention in Saratchandran et al. (2024). In addition, Aliakbarpour et al. (2025) showed that under certain structural properties, softmax attention can be efficiently simulated using a combination of polynomial attentions and sketching methods. Motivated by these positive results, we study the polynomial sampler that reports an index with probability $\frac{|(Ax)_i|^2}{\sum_{j=1}^n |(Ax)_j|^2}$. We obtain fast and space-efficient polynomial attention samplers under various settings, summarized as follows:

**Theorem 1.3.** *Given a matrix $X \in \mathbb{R}^{n^2 \times d^2}$, a vector $w \in \mathbb{R}^{d^2}$, and an accuracy parameter $\epsilon$, then there exists algorithms that sample an index $i \in [n]$ with probability proportional to $p_i = (1 \pm \epsilon) \cdot \frac{|(Ax)_i|^2}{\sum_{j=1}^n |(Ax)_j|^2}$ at all times in the following settings:*

- *Updating $A$ and fixed $x$: the algorithm uses $d^2 \log n + \text{poly}\left(\frac{1}{\epsilon}, \log n\right)$ bits of space and $\text{poly}\left(\frac{1}{\epsilon}, \log n\right)$ update time (see Theorem 5.3).*

- *Fixed $A$ and updating $x$: the algorithm uses $d^2 \text{poly}\left(\frac{1}{\epsilon}, \log n\right)$ bits of space and $O(1)$ update time (see Theorem 5.5).*

- *Updating both $A$ and $x$: the algorithm uses $d^2 \text{poly}\left(\frac{1}{\epsilon}, \log n\right)$ bits of space and $\text{poly}\left(\frac{1}{\epsilon}, \log n\right)$ update time (see Theorem 5.7).*

*In addition, for updating both $A$ and $x$, any polynomial attention sampler requires $\Omega(d^2)$ space (see Theorem 6.2).*

Our results establish the first algorithmic framework for sampling polynomial attention mechanisms with space complexity that is strictly sublinear in the sequence length $n$, bypassing the quadratic barrier in sampling and estimating softmax attention. This efficiency is particularly notable in turnstile streams, where our algorithms handle dynamic updates to both input tokens and model weights. We complement our results by showing the space lower bound when updating both $A$ and $x$, which shows that our space complexity is tight up to poly-logarithmic factors.

Finally, we provide attention samplers for the tensor formulation of the attention matrix $(A_1 \otimes A_2)x$, where $x$ is fixed and one of $A_1$ and $A_2$ is updating. We note that the algorithm has faster update time compared to explicitly updating $A = A_1 \otimes A_2$ and applying the algorithms in Theorem 1.3.

**Theorem 1.4.** *[Informal statement of Theorem 7.6] Given matrices $A_1, A_2 \in \mathbb{R}^{n \times d}$ for $A = A_1 \otimes A_2 \in \mathbb{R}^{n^2 \times d^2}$, where one of $A_1$ and $A_2$ is updating, and fixed $x \in \mathbb{R}^{d^2}$, we sample $(i_1, i_2) = i \in [n^2]$ approximately according to the $\ell_2$ sampling distribution on $Ax \in \mathbb{R}^{n^2}$. The algorithm uses $O(nd)$ space and $O(n)$ update time. Note that the trivial result takes $O(n^2)$ space.*

## 2 RELATED WORK

In this section, we present related work in sampling and tensor sketch.

**On sampling.** Given a vector $v \in \mathbb{R}^n$ and a distribution function $g$, recall that the classical $g$-sampler samples index $i \in [n]$ with probability $p_i = \frac{g(v_i)}{\sum_{j=1}^n g(v_j)}$. A well-known example is the $L_p$ sampling defined by $g(z) = |z|^p$ for $p \geq 0$. The existence of such a $L_p$ sampler algorithms first posed as a question by Cormode et al. (2005) in 2005. Monemizadeh & Woodruff (2010) partially answered this question in the affirmative by giving an $L_p$ sampler using polylogarithmic space for $p \in [1, 2]$, although the sampling probabilities were distorted by a multiplicative $(1 + \epsilon)$ factor and an additive $\frac{1}{\text{poly}(n)}$ factor. We note that the sampler is perfect if there is no $\epsilon$-multiplicative distortion; it is truly perfect if there is no additive distortion, i.e., the sampling probability is exact. The space requirements of the algorithm were subsequently improved (Andoni et al., 2011; Jowhari et al., 2011) and extended to other choices of index domain $U$ and weight function $W$ (Cohen & Geri, 2019; Mahabadi et al., 2020; 2022), while retaining a multiplicative distortion in the sampling probability. Surprisingly, Jayaram & Woodruff (2021) showed that it is possible to achieve no perfect samplers while using polylogarithmic space, while conversely Jayaram et al. (2022) showed that truly perfect samplers would require linear space, essentially closing the line of work studying the space complexity of $L_p$ samplers for $p \in [1, 2]$. It should be noted however, achieving such guarantees (no additive distortion) in sub-polynomial update time while retaining the space guarantees remains an intriguing open question (Jayaram et al., 2022). For the other regime of $p > 2$, recently, Woodruff et al. (2025) complemented the results by providing efficient perfect $L_p$ samplers for $p > 2$. Swartworth et al. (2025) achieved perfect samplers with polylogarithmic update time for $p > 2$, improving on the previous update time. For a more comprehensive background on samplers, we refer to the survey by Cormode & Jowhari (2019).

**On tensors.** In the realm of tensor decomposition, the canonical polyadic (CP) decomposition, specifically the CANDECOMP/PARAFAC method, stands out for its unique ability to break down tensors into rank-1 tensors in a singular way, distinct from matrix decomposition (Harshman, 1970; Song et al., 2016). This method, having applications in computational neuroscience, data mining, and statistical learning (Wang et al., 2015), emphasizes the rigidity and uniqueness of tensor decomposition. Earlier studies (Tsourakakis, 2010; Phan et al., 2013; Choi & Vishwanathan, 2014; Huang et al., 2013; Kang et al., 2012; Wang et al., 2014; Bhojanapalli & Sanghavi, 2015) have delved into efficient tensor decomposition methods. Subsequent works introduced methods for fast orthogonal tensor decomposition using random linear sketching techniques (Wang et al., 2015) and explored symmetric orthogonally decomposable tensors' properties, integrating spectral theory (Robeva, 2016; Robeva & Seigal, 2017). Additionally, importance sampling for quicker decomposition was proposed (Song et al., 2016). (Deng et al., 2023a) studies the tensor cycle low rank approximation problem.

In algebraic statistics, tensor decompositions are linked to probabilistic models, particularly in determining latent variable models' identifiability through low-rank decompositions of specific moment tensors (Allman et al., 2009a;b; Rhodes & Sullivant, 2012). Kruskal's theorem (Kruskal, 1977) was pivotal in ascertaining the precision of model parameter identification. However, this approach, assuming an infinite sample size, does not provide the minimum sample size for learning model parameters within given error bounds. A more robust uniqueness guarantee is needed to ensure that the low-rank decomposition of an empirical moment tensor approximates that of an actual moment tensor, thus offering more insight into empirical moment tensors' decomposition.

**Roadmap.** In Section 3, we provide some standard notations and definitions in literature. In Section 4, we study the exponential sampler. In Section 5, we study the streaming upper for the $\ell_2$ sampling problem, i.e., sampling coordinates from a vector $Ax$, where $A$ and $x$ may be updated across a data stream. In Section 6, we present lower bounds for the same $\ell_2$ sampling problem. In Section 7, we discuss the tensor sampling problem.

## 3 PRELIMINARIES

For any positive integer $n$, we use $[n]$ to denote the set $\{1, 2, \cdots, n\}$. We use $\mathbb{E}[\cdot]$ to denote the expectation. We use $\Pr[\cdot]$ to denote the probability. We use $\mathbf{1}_n$ to denote a length-$n$ vector where all the entries are ones. Given two length-$n$ vectors, we use $\langle x, y \rangle$ to denote the inner product

between $x$ and $y$, i.e, $\langle x, y \rangle := \sum_{i=1}^n x_i y_i$. For a vector $x \in \mathbb{R}^n$, we use $\exp(x) \in \mathbb{R}^n$ to denote a vector that has length $n$ and the $i$-th entry is $\exp(x_i)$. For a matrix $A$, we use $\exp(A)$ to denote the matrix that $(i,j)$-th coordinate is $\exp(A_{i,j})$. For a vector $x$, we use $\|x\|_2 := (\sum_{i=1}^n x_i^2)^{1/2}$. We use $\|x\|_1 := \sum_{i=1}^n |x_i|$. We use $\|x\|_0$ to denote the $\ell_0$ norm of $x$, which is the number of nonzero entries in $x$. We use $\|x\|_\infty$ to denote the $\ell_\infty$ norm of $x$, which is $\max_{i \in [n]} |x_i|$.

Let $n_1, n_2, d_1, d_2$ be positive integers. Let $A \in \mathbb{R}^{n_1 \times d_1}$ and $B \in \mathbb{R}^{n_2 \times d_2}$. We define the Kronecker product between matrices $A$ and $B$, denoted $A \otimes B \in \mathbb{R}^{n_1 n_2 \times d_1 d_2}$, as $(A \otimes B)_{(i_1-1)n_2+i_2, (j_1-1)d_2+j_2}$ is equal to $A_{i_1, j_1} B_{i_2, j_2}$, where $i_1 \in [n_1], j_1 \in [d_1], i_2 \in [n_2], j_2 \in [d_2]$.

We use $\mathrm{poly}(n)$ to denote $n^C$ where $C > 1$ is some constant. For any function $f$, we use $\widetilde{O}(f)$ to denote $f \cdot \mathrm{poly}(\log f)$. For two sets $A$ and $B$, we use $A \cap B$ to denote their intersection. We use $|A \cap B|$ to denote the cardinality of $A \cap B$. We use $A \cup B$ to denote the union of $A$ and $B$.

**TensorSketch.** We next define TensorSketch (Pagh, 2013), which has been extensively used in many sketching and optimization problems (Diao et al., 2018; Song et al., 2019; Diao et al., 2019; Ahle et al., 2020; Song et al., 2021; 2024; 2022; Zhang, 2022; Song et al., 2023b). Song et al. (2022) defined TensorSparse by composing Sparse embedding (Nelson & Nguyên, 2013; Cohen, 2016) with a tensor operation (Pagh, 2013).

**Definition 3.1** (TensorSparse, see Definition 7.6 in Song et al. (2022))**.** Let $h_1, h_2 : [n] \times [s] \to [m/s]$ be $O(\log 1/\delta)$-wise independent hash functions and let $\sigma_1, \sigma_2 : [n] \times [s] \to \{\pm 1\}$ be $O(\log 1/\delta)$-wise independent random sign functions. Then, the degree two tensor sparse transform, $S : \mathbb{R}^n \times \mathbb{R}^n \to \mathbb{R}^m$ is given as:

$$R_{r,(i,j)} = \exists k \in [s] : \sigma_1(i,k)\sigma_2(j,k)/\sqrt{s} \cdot \mathbf{1}[((h_1(i,k) + h_2(j,k)) \bmod m/s) + (k-1)m/s = r]$$

For $s = 1$, the above definition becomes TensorSketch (Pagh, 2013).

# 4 EXPONENTIAL SAMPLER

In this section, we define and consider exponential samplers. We then show strong space lower bounds for achieving such a data structure when the input dataset arrives in a data stream. We remark that we let $A$ be an $n^2 \times d^2$ matrix when we define the attention matrix in the introduction (see e.g., Definition 1.1), since $A$ is the tensor product $A_1 \otimes A_2$ of two input matrices $A_1, A_2 \in \mathbb{R}^{n \times d}$. However, in the following sections, for simplicity of expression, we let $A$ be an $n \times d$ matrix, and hence the goal is to design attention samplers with space sublinear in $n$.

Let us firstly describe the offline version:

**Definition 4.1** (Exponential sampler)**.** Given matrix $A \in \mathbb{R}^{n \times d}$ and $x \in \mathbb{R}^d$, the goal is to sample index $i \sim [n]$ with probability $p_i = \langle \exp(Ax), \mathbf{1}_n \rangle^{-1} \cdot \exp(Ax)_i$, where $\mathbf{1}_n$ denotes a length-$n$ vector, $\exp(Ax) \in \mathbb{R}^n$ denotes a length-$n$ vector with $\exp(Ax)_i = \exp((Ax)_i)$, and $\exp(z)$ is the usual exponential function.

Now, consider $y = Ax \in \mathbb{R}^n$, where either $A$ or $x$, or both are arriving in a data stream, we use the following definition for each of the various cases:

**Definition 4.2.** Let $C > 0$ be any fixed constant and let $C_0 \in [n^{-C}, n^C]$. Let $y$ be a vector. Then the exponential sampler outputs an index $j^*$ such that for all $i \in [n]$, $\Pr[j^* = i] = C_0 \cdot \frac{\exp(y_i)}{\langle \exp(y), \mathbf{1}_n \rangle}$.

We first recall the (two-party) set-disjointness communication problem $\mathsf{SetDisj}_n$, in which two parties Alice and Bob have subsets $A$ and $B$, respectively, of $[n]$. Note that we can equivalently view $A$ and $B$ as binary vectors in $n$-dimensional space, serving as the indicator vector for whether each index $i \in [n]$ is in the player's input subset. The task for the players is to determine whether there exists a common element in their intersection, i.e., whether there exists $i \in [n]$ such that $i \in (A \cap B)$ or equivalently, $A_i = B_i = 1$. In fact, the problem promises that either the inputs are completely disjoint, $|A \cap B| = 0$ or the inputs contain only a single coordinate in their intersection, $|A \cap B| = 1$. We recall the following standard communication complexity result of set-disjointness.

**Theorem 4.3** (Kalyanasundaram & Schnitger (1992); Razborov (1992); Bar-Yossef et al. (2004))**.** *Any protocol that solves the set-disjointness problem* $\mathsf{SetDisj}_n$ *with probability at least* $\frac{3}{4}$ *requires* $\Omega(n)$ *bits of total communication.*

We show that even a sampler that relaxes the probability distribution defined in Definition 4.2 up to a factor of $n^C$ is infeasible in the streaming model.

**Theorem 4.4.** *Let $y \in \mathbb{R}^n$ that arrives as a data stream and let $C > 0$ be a constant. Then any algorithm that samples an index $i \in [n]$ with probability proportional to $p_i = \frac{\exp(y_j)}{\langle \exp(y), \mathbf{1}_n \rangle}$ must use $\Omega(n)$ bits of space, even if the sampling probabilities are allowed to be distorted by as large as $n^C$ and even if $\|y\|_\infty = O(\log n)$.*

*Proof.* Let $A, B \in \{0,1\}^n$ be input vectors from the set disjointness problem, so that the goal is to determine whether there exists $i \in [n]$ such that $A_i = B_i = 0$. Observe that Alice and Bob can multiply $A$ and $B$ by $100C \log n$ for some constant $C > 0$. Now, note that in the disjoint case, we have that $\|A + B\|_\infty = 100C \log n$ and in the non-disjoint case, we have that $\|A + B\|_\infty = 200C \log n$. In particular, in the non-disjoint case, there exists $i \in [n]$ such that $A_i + B_i = 200C \log n$ and for all $j \neq i$, we have that $A_j + B_j \leq 100C \log n$. Hence, in the non-disjoint case, any exponential sampler will output $i$ with probability proportional to $\exp(200C \log n)$ and output $j \neq i$ with probability proportional to $n \cdot \exp(100C \log n)$. Even if the sampling probabilities are distorted by a factor of $n^C$, any exponential sampler would output $i$ with probability at least $\frac{3}{4}$.

Thus, Alice and Bob can use such a data structure to sample an index $i$ and then check whether $A_i = B_i = 1$. In particular, Alice can first create a data stream encoding the vector $A$, run the sampling algorithm on the data stream, and then pass the state of the algorithm to Bob. Bob can then create another portion of the data stream encoding an addition of the vector $B$, take the state of the algorithm from Alice, run the sampling algorithm on the portion of the data stream, and query the algorithm for an index $i$. Bob can then take the index and pass it to Alice, and the two parties can finally communicate whether $A_i = B_i = 1$, thereby solving set-disjointness with probability at least $\frac{3}{4}$. Note that the communication of the protocol is the space used by the sampling algorithm. Therefore by Theorem 4.3, such a sampler must use $\Omega(n)$ bits of space. $\square$

## 5 $\ell_2$ SAMPLER UPPER BOUND WITH $A$ AND $x$

In this section, we describe a standard data structure for $\ell_2$ sampling. We start with providing the definition of $\ell_2$ sampler as follows,

**Definition 5.1.** Let $n$ denote a positive integer. Let $\epsilon \geq 0$ denote a parameter. In $\ell_2$ sampling, we receive each coordinate of $y \in \mathbb{R}^n$ in a turnstile data stream, and the goal is to output an index $I \in [n]$ at all times such that for each $j \in [n]$, $\Pr[I = j] = (1 \pm \epsilon) \cdot \frac{|y_j|^2}{\|y\|_2^2} + 1/\operatorname{poly}(n)$.

We describe various instantiations of the $\ell_2$ sampler for sampling entries from a vector $Ax \in \mathbb{R}^n$, based upon whether the matrix $A \in \mathbb{R}^{n \times d}$ is updated during the data stream, whether the vector $x \in \mathbb{R}^d$ is updated during the data stream, or both.

### 5.1 $A$ IS UPDATED DURING THE STREAMING AND $x$ IS FIXED

In this section, we describe the construction of an $\ell_2$ sampler for sampling coordinates of the vector $Ax \in \mathbb{R}^n$, in the setting where the vector $x \in \mathbb{R}^d$ is fixed, but the entries of $A \in \mathbb{R}^{n \times d}$ are evolving as the data stream progresses.

**Definition 5.2** (Updating $A$ and fixed $x$)**.** In this setting, we assume $x \in \mathbb{R}^d$ is fixed, we receive updates to the entries of $A \in \mathbb{R}^{n \times d}$ in a turnstile data stream. Then for $y = Ax$, we want a data structure that produces the $\ell_2$ sampling guarantee for $y$.

We remark that a turnstile data stream means that each update of the data stream can increase or decrease a single entry of $A$.

In this work, we are interested in the regime of $n \gg d$. Then we have the following guarantee:

**Theorem 5.3.** *Suppose $y = Ax$, for $x \in \mathbb{R}^n$, which is fixed, and $A \in \mathbb{R}^{n \times d}$, which is defined by a turnstile stream. There exists an $\ell_2$-attention sampler that uses $d \log n + \operatorname{poly}\left(\frac{1}{\epsilon}, \log n\right)$ bits of space and returns $I \in [n]$ such that $\Pr[I = j] = (1 \pm \epsilon) \cdot \frac{|y_j|^2}{\|y\|_2^2} + 1/\operatorname{poly}(n)$. The update time of the data structure is $\operatorname{poly}\left(\frac{1}{\epsilon}, \log n\right)$.*

*Proof.* Recall that existing approximate $\ell_2$ samplers, e.g., Algorithm 2 maintains a linear sketch $\Phi y$, where $\Phi \in \mathbb{R}^{m \times n}$, for $m = \text{poly}\left(\frac{1}{\epsilon}, \log n\right)$. We have $y = Ax$, where $x \in \mathbb{R}^d$ is fixed but $A \in \mathbb{R}^{n \times d}$ is defined through turnstile updates. Nevertheless, we can maintain the state of $\Phi Ax$. In particular, whenever we receive an update in $A_{i,j}$ by $\Delta$, then we can compute $\Phi e_i e_j^\top \Delta x$ to update the sketch $\Phi Ax$. To analyze the space complexity, observe that storing $\Phi Ax$ requires $O(m)$ words of space and $x$ requires $d$ words of space, which is $d \log n + \text{poly}\left(\frac{1}{\epsilon}, \log n\right)$ bits of space in total. Moreover, we need to update all the $m$ entries in $\Phi Ax$ upon each update, using $O(m) = \text{poly}\left(\frac{1}{\epsilon}, \log n\right)$ time.

$\square$

## 5.2 $x$ IS UPDATED DURING THE STREAMING AND $A$ IS FIXED

We next consider the setting where the vector $x \in \mathbb{R}^d$ is updated as the data stream progresses, but the entries of $A \in \mathbb{R}^{n \times d}$ are fixed.

**Definition 5.4** (Fixed $A$ and updating $x$). We assume $A \in \mathbb{R}^{n \times d}$ is fixed, we receive updates to $x \in \mathbb{R}^d$ in a turnstile data stream. Then for $y = Ax$, we want a data structure that produces the $\ell_2$ sampling guarantee for $y$.

We have the following algorithmic guarantees for this setting:

**Theorem 5.5.** *Suppose $y = Ax$, for $A \in \mathbb{R}^{n \times d}$, which is fixed, and $x \in \mathbb{R}^n$, which is defined by a turnstile stream. There is an $\ell_2$-attention sampler that uses $d \, \text{poly}\left(\frac{1}{\epsilon}, \log n\right)$ bits of space and returns $I \in [n]$ such that $\Pr[I = j] = (1 \pm \epsilon) \cdot \frac{|y_j|^2}{\|y\|_2^2} + 1/\text{poly}(n)$. The update time of the data structure is $O(1)$.*

*Proof.* Again recall that existing approximate $\ell_2$ samplers, e.g., Algorithm 2 maintains a linear sketch $\Phi y$, where $\Phi \in \mathbb{R}^{m \times n}$, for $m = \text{poly}\left(\frac{1}{\epsilon}, \log n\right)$. Since $y = Ax$, but $A \in \mathbb{R}^{n \times d}$ is too large to store, while $x \in \mathbb{R}^n$ is defined through turnstile updates, we can instead maintain the sketch $\Phi A$ and the vector $x$ and compute $\Phi Ax = \Phi y$ after the stream concludes. Note that storing $\Phi A$ requires $O(md)$ words of space and $x$ requires $d$ words of space, which is $d \, \text{poly}\left(\frac{1}{\epsilon}, \log n\right)$ bits of space in total. Moreover, each update to $x$ changes a single entry, so the update time is $O(1)$. $\square$

## 5.3 BOTH $A$ AND $x$ ARE UPDATED DURING THE STREAMING

Finally, we consider the setting where both the vector $x \in \mathbb{R}^d$ and the entries of $A \in \mathbb{R}^{n \times d}$ can be changed by updates from the data stream.

**Definition 5.6** (Updating $A$ and updating $x$). In this setting, we receive updates to both $A \in \mathbb{R}^{n \times d}$ and $x \in \mathbb{R}^d$ in a turnstile data stream. Then for $y = Ax$, we want a data structure that provides the $\ell_2$ sampling guarantee for $y$.

We have the following guarantees:

**Lemma 5.7** (Upper Bound). *Suppose $y = Ax$, for $A \in \mathbb{R}^{n \times d}$ and $x \in \mathbb{R}^d$, which are each defined in a stream through turnstile updates. There exists an $\ell - 2$-attention sampler that uses $d \, \text{poly}\left(\frac{1}{\epsilon}, \log n\right)$ bits of space and returns $I \in [n]$ such that $\Pr[I = j] = (1 \pm \epsilon) \cdot \frac{|y_j|^2}{\|y\|_2^2} + 1/\text{poly}(n)$. The update time is $\text{poly}\left(\frac{1}{\epsilon}, \log n\right)$.*

*Proof.* As before, recall that existing approximate $\ell_2$ samplers, e.g., Algorithm 2 maintains a linear sketch $\Phi y$, where $\Phi \in \mathbb{R}^{m \times n}$, for $m = \text{poly}\left(\frac{1}{\epsilon}, \log n\right)$. Since $y = Ax$, but now both $A \in \mathbb{R}^{n \times d}$ and $x \in \mathbb{R}^n$ are defined through turnstile updates, we can instead maintain the sketch $\Phi A$ and the vector $x$ and compute $\Phi Ax = \Phi y$ after the stream concludes. Observe that maintaining $\Phi A$ requires $O(md)$ words of space and $x$ requires $d$ words of space, which is $d \, \text{poly}\left(\frac{1}{\epsilon}, \log n\right)$ bits of space in total. Each update to $A$ can change all $m$ entries of in a single column of $\Phi A$, while each update to $x$ changes a single entry. Hence, the update time is $\text{poly}\left(\frac{1}{\epsilon}, \log n\right)$. $\square$

# 6  $\ell_2$ SAMPLER LOWER BOUND (WITH $A$ AND $x$)

In this section, we give lower bounds for $\ell_2$ sampling from a vector $y = A^{\otimes p}x$, when either $A$ or $x$ are updated in a data stream. We show that in any of these cases, the general problem is substantially more difficult than the previous case where $p = 1$.

We first recall the Index problem for one-way communication. In the $\mathsf{INDEX}_n$ problem, Alice receives a vector $v \in \{0, 1\}^n$ and Bob receives a coordinate $i \in [n]$. The goal is for Bob to compute $v_i$ with probability at least $\frac{3}{4}$, given some message $\Pi$ from Alice. We recall the following communication complexity lower bounds for Index.

**Theorem 6.1** (Kremer et al. (1999)). *Any protocol that solves $\mathsf{INDEX}_n$ with probability at least $\frac{3}{4}$ requires $\Omega(n)$ bits of communication.*

**Lemma 6.2** (Lower Bound). *Any streaming algorithm that solves problem defined as Definition 5.6 will require $\Omega(d)$ space.*

*Proof.* Suppose Alice receives a vector $v \in \{0, 1\}^d$. Then Alice creates the diagonal matrix $M \in \{0, 1\}^{d \times d}$ so that the $j$-th diagonal entry of $A$ is $v_j$, for all $j \in [n]$. Finally, Alice creates $A \in \mathbb{R}^{(d+1) \times d}$ by appending the row consisting of $\frac{1}{10^{10}}$ in all of its $d$ entries to $M$. Suppose Bob receives the coordinate $i \in [d]$ and wants to determine $v_i$. Then Bob can set $x$ to be the elementary vector $e_i \in \mathbb{R}^d$, which has a 1 in its $i$-th coordinate and zeros elsewhere. Observe that by construction, $Ax$ is the $i$-th column of $A$. If $v_i = 1$, then the $i$-th column of $A$ consists of a 1 in the $i$-th entry, $\frac{1}{10^{10}}$ in the $(d+1)$-st entry, and zeros elsewhere. Hence, a sampler with the desired properties will output $i$ with probability at least $\frac{3}{4}$. Similarly, if $v_i = 0$, then the $i$-th column of $A$ consists of $\frac{1}{10^{10}}$ in the $(d+1)$-st entry and zeros elsewhere. Thus, the sampler with the desired properties will output $d+1$ with probability 1. Bob can therefore distinguish between these two cases with probability at least $\frac{3}{4}$, thereby solving $\mathsf{INDEX}_d$ with probability at least $\frac{3}{4}$. Therefore, by Theorem 6.1, such a sampler must use at least $\Omega(d)$ space. $\qquad\square$

In fact, we show that if $y = A^{\otimes p}x$, where $A \in \mathbb{R}^{n \times n}$ so that $A^{\otimes p} \in \mathbb{R}^{n^p \times n^p}$ denotes the $p$-wise self-tensor and $x \in \mathbb{R}^{n^p}$, then actually $L_2$ sampling from $y$ uses $\Omega(n)$ bits of space.

**Lemma 6.3.** *Let $A \in \mathbb{R}^{n \times n}$ and $A^{\otimes p} \in \mathbb{R}^{n^p \times n^p}$ denote the $p$-wise self-tensor. Let $y = A^{\otimes p}x$, so that $x \in \mathbb{R}^{n^p}$. Then even if all the entries of $x$ arrive in a data stream followed by all the entries of $A$, $L_2$ sampling from $y$ requires $\Omega(n)$ bits of space.*

*Proof.* Let $S \in \{0, 1\}^n$ be an instance of $\mathsf{INDEX}_n$. Suppose Alice creates the diagonal matrix $A$ with exactly $S$ being the entries across its diagonal, i.e., $A_{1,1} = S_1, \ldots, A_{n,n} = S_n$. Bob has an index $i \in [n]$, and sets the vector $x$ to be the elementary vector $\mathbf{e}_j$, where $j = i \cdot n^{p-1}$. Then by construction $Ax$ is the all zeros vector if $S_i = 0$ and otherwise there is a nonzero entry, which allows Alice and Bob to solve $\mathsf{INDEX}_n$. Hence, $L_2$ sampling from $y$ requires $\Omega(n)$ bits of space. $\qquad\square$

# 7  THE TENSOR VERSION PROBLEM

In this section, we further consider sampling from a tensor product. We provide the tensor notations and objects.

**Definition 7.1.** Let $A_1 \in \mathbb{R}^{n \times d}$, let $A_2 \in \mathbb{R}^{n \times d}$, we define $\mathsf{A} = A_1 \otimes A_2 \in \mathbb{R}^{n^2 \times d^2}$. Let $x \in \mathbb{R}^{d^2}$. Let $\mathsf{A}_i \in \mathbb{R}^{n \times d^2}$ denote the $i$-th block of $\mathsf{A}$.

**Definition 7.2** (Fixed $x$, streaming sampler for one of $A_1$ and $A_2$ is updating.). We assume $x \in \mathbb{R}^{d^2}$ is fixed. We assume that (1) one of $A_1$ and $A_2$ is updating, (2) one of $A_1$ and $A_2$ is fixed. Let $y = \mathsf{A}x$, we want $\ell_2$ sampling guarantee for sampling one coordinate in $y_i \in \mathbb{R}^{n^2}$ for all $i \in [n^2]$.

To motive this model, recall that the tensor product $(A_1 \otimes A_2)x$ equals to the linear cross-attention matrix $QK^\top = A_1 W_Q W_K^\top A_2^\top$, where $Q = A_1 W_Q$ is the query matrix and $K = A_2 W_K$ is the key matrix. Our model addresses a practical scenario involving real-time contextual processing with a static reference dataset. In this setting, the key matrix $K$ is precomputed by the language model, representing a static dataset such as embeddings of a knowledge base, user profiles, or multimedia

features. Then, the entries of input matrix $A_1$ is updated by a turnstile data stream, which changes the entries in the query matrix $Q$ implicitly, representing real-time dynamic data queries. In particular, our model is suitable for auto-regressive Transformers, where the key matrix $K$ is given and the rows of input matrix $A_1$ arrives one-by-one as a data stream. Thus, our attention sampler recovers the important coordinates in attention computation in auto-regressive Transformers.

We use the following formulation of Nisan's pseudorandom generator to derandomize our algorithm.

**Theorem 7.3** (Nisan's PRG, Nisan (1992)). *Suppose $\mathcal{A}$ is an algorithm that requires $S = \Omega(\log n)$ bits of space and $R$ random bits. Then there exists a pseudorandom generator for $\mathcal{A}$ that succeeds with probability $1 - 1/\mathrm{poly}(n)$ and uses $O(S \log R)$ bits of space.*

---

**Algorithm 1** Algorithm for sampling based on $S(x_1 \otimes x_2)$

---

1: Suppose we use $O(nd)$ space to store $A_1$ and $A_2$ (Avoid $n^2$ time/space)
2: Suppose we receive an update $q \in [2]$, $i \in [n]$, $j \in [d]$, $\Delta$
3: Suppose we have hash function $g$ to access uniform number
4: **if** $q = 1$ **then**
5: $\quad p \leftarrow g(i(n-1)+1, \cdots, in)$ $\{p \in \mathbb{R}^n\}$
6: $\quad y \leftarrow y + \Phi\Delta(e_{[i(n-1)+1,in]} \circ (A_2)_{*,j})/p$ $\{\Phi_1 \text{ is decided by } h_1, \sigma_1\}$
7: **else**
8: $\quad y_2 \leftarrow y_2 + \Phi_2 e_i \Delta$ $\{\Phi_2 \text{ is decided by } h_2, \sigma_2\}$
9: **end if**

---

In the following Lemma, we state a streaming algorithm to solve tensor related sampling problem. We consider the situation that one of $A_1$ and $A_2$ is fixed, and the other one is updated in streaming fashion. We show the following estimation guarantees using the standard CountSketch analysis, c.f., Charikar et al. (2004); Jowhari et al. (2011). We defer the proof to appendix C.

**Lemma 7.4** (Tensor $\ell_2$ Tail Estimation). *Let $y = (A_1 \otimes A_2)x \in \mathbb{R}^{n^2}$. Let only one of $A_1$ and $A_2$ be updated in streaming. Let $w = \frac{y_i}{\sqrt{u_i}}$ for a constant $u_i \in [0,1]$ generated uniformly at random. There is an algorithm $\mathcal{A}$ that that uses $O(nd) + \mathrm{poly}\left(\frac{1}{\epsilon}, \log n\right)$ space, uses $O(n)$ update time, and estimates each element of $w$ up to additive error $\epsilon \cdot \|z\|_2$, where $z$ denotes the tail vector of $w$ without the largest $\frac{1}{\epsilon^2}$ entries in magnitude. Specifically, for all $i \in [n^2]$, we have $|\widehat{w}_i - w_i| \leq \epsilon \cdot \|z\|_2$.*

We state the following lemma as a structural property that will allow us to achieve our tensor product sampler. We remark that the proof is a simple adaptation of existing proofs for approximate $\ell_p$ sampling (Jowhari et al., 2011). Thus we defer the proof to Appendix C.

**Lemma 7.5.** *Let $y = (A_1 \otimes A_2)x \in \mathbb{R}^{n^2}$ and let $w \in \mathbb{R}^{n^2}$ so that $w_i = \frac{y_i}{\sqrt{u_i}}$ for a constant $u_i \in [0,1]$ generated uniformly at random. Let $z$ denote the tail vector of $w$ without the largest $\frac{1}{\epsilon^2}$ entries in magnitude. Let $\widehat{Z}$ be a 2-approximation to $\|z\|_2$ and $\widehat{Y}$ be a 2-approximation to $\|y\|_2$, then we have $\Pr[\widehat{Z} > \sqrt{(C \log n)/\epsilon} \cdot \widehat{Y}] \leq O(\epsilon) + \frac{1}{\mathrm{poly}(n)}$.*

Finally, we describe the guarantees of our tensor-based sampler, deferring the proof to Appendix D.

**Theorem 7.6.** *Let $y = (A_1 \otimes A_2)x \in \mathbb{R}^{n^2}$ and let $w \in \mathbb{R}^{n^2}$ so that for each $i \in [n^2]$, $w_i = \frac{y_i}{\sqrt{u_i}}$ for a constant $u_i \in [0,1]$ generated uniformly at random. Let $z$ denote the tail vector of $w$ without the largest $\frac{1}{\epsilon^2}$ entries in magnitude. Suppose there exists:*

1. *An algorithm $\mathcal{A}_1$ that provides a 2-approximation to $\|y\|_2$ with probability $1 - \frac{1}{n^2}$.*

2. *An algorithm $\mathcal{A}_2$ that provides a 2-approximation to $\|z\|_2$ with probability $1 - \frac{1}{n^2}$.*

3. *An algorithm $\mathcal{A}_3$ that estimates each element of $w$ up to additive error $\epsilon \cdot \|z\|_2$, $|\widehat{w}_i - w_i| \leq \epsilon \cdot \|z\|_2$, for all $i \in [n^2]$.*

*Then there exists a data structure that uses $\mathrm{poly}\left(\frac{1}{\epsilon}, \log n\right)$ bits of space, $O(n)$ update time, and outputs each index $i$ with probability $p_i = (1 \pm \epsilon) \cdot \frac{y_i^2}{\|y\|_2^2} \pm \frac{1}{\mathrm{poly}(n)}$.*

We remark that the algorithms $\mathcal{A}_1$ and $\mathcal{A}_2$ in the context of Theorem 7.6 can be achieved using the standard AMS $\ell_2$ norm estimator (Alon et al., 1999). Moreover, algorithm $\mathcal{A}_3$ in the context of Theorem 7.6 can be achieved using the standard CountSketch algorithm (Charikar et al., 2004).

## 8 CONCLUSIONS

To achieve efficient attention mechanisms, we introduce the attention sampler and study its behavior in the streaming model. We established efficient polynomial samplers under various streaming settings, when the input matrix, the weight vector, or both evolve dynamically, and we complement the results by proving space lower bounds. Our sampling framework recovers the important coordinates in the attention matrix, offering a foundation for efficient simulations of large-scale attention schemes, which is central to modern machine learning and LLMs.

For future directions, from a theoretical perspective, given the $\Omega(n)$ lower bound on exponential samplers in general circumstances, it would be valuable to explore whether we can achieve $o(n)$ space under certain assumptions, e.g., restricting the entries in the attention matrix to $o(\log n)$. From a practical perspective, it would be beneficial to evaluate our sampler's performance in real-world Transformer applications, especially in sparse attention schemes and streaming attention schemes.

## ACKNOWLEDGMENTS

Shenghao Xie is supported in part by NSF CCF-2335411. Samson Zhou is supported in part by NSF CCF-2335411. Samson Zhou gratefully acknowledges funding provided by the Oak Ridge Associated Universities (ORAU) Ralph E. Powe Junior Faculty Enhancement Award.

## ETHIC STATEMENT

This paper does not involve human subjects, personally identifiable data, or sensitive applications. We do not foresee direct ethical risks. We follow the ICLR Code of Ethics and affirm that all aspects of this research comply with the principles of fairness, transparency, and integrity.

## REPRODUCIBILITY STATEMENT

We ensure reproducibility of our theoretical results by including all formal assumptions, definitions, and complete proofs in the appendix. The main text states each theorem clearly and refers to the detailed proofs. No external data or software is required.

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

# Appendix

**Roadmap.** In Section A, we provide additional related work. In Section B, we briefly discuss the background on $\ell_2$ sampler. In Section C, we show that how to use the tail bound to obtain sampling result. In Section D, we present the tensor sampling result. In Section E, we discuss the LLM usage of the paper.

## A ADDITIONAL RELATED WORK

**On sketching.**

The application of sketching and sampling techniques in numerical linear algebra has been remarkably effective, revolutionizing a broad spectrum of core tasks. These methods are crucial in linear programming (LP), as evidenced by Cohen et al. (2019); Jiang et al. (2021); Ye (2020); Gu & Song (2022), and have significantly impacted tensor approximation (Song et al., 2019; Mahankali et al., 2024; Deng et al., 2023a). Sketching and sampling techniques also have been widely applied in matrix completion (Gu et al., 2024), matrix sensing (Qin et al., 2024; Deng et al., 2023b), submodular function maximization (Qin et al., 2023a), dynamic sparsification (Deng et al., 2022a), dynamic tensor product regression (Reddy et al., 2022), and semi-definite programming (Song et al., 2023a). Additionally, sketching has been pivotal in iterative sparsification problems (Song et al., 2022), adversarial training (Gao et al., 2022), kernel density estimation (Qin et al., 2022b), solving the distance oracle problem (Deng et al., 2022b), and empirical risk minimization (Lee et al., 2019; Qin et al., 2023b). Its applications furthermore extends to relational databases (Qin et al., 2022a) and Large Language Model (LLM) research (Deng et al., 2023c;b; Gao et al., 2025; Li et al., 2023).

**On theoretical attention.**

A comprehensive body of research, including studies (Child et al., 2019; Kitaev et al., 2020; Wang et al., 2020; Daras et al., 2020; Katharopoulos et al., 2020; Chen et al., 2021; 2022; Zandieh et al., 2023; Alman & Song, 2023; Brand et al., 2024; Deng et al., 2023c; Kacham et al., 2023; Alman & Song, 2024a; Han et al., 2024; Awasthi & Gupta, 2023; Marcus et al., 2022; Alman & Song, 2024b; 2025a;b), has progressively shed light on the complexities and optimization of attention matrix computation. This exploration has been further enriched by insights into the effectiveness of attention mechanisms in Transformers (Dehghani et al., 2018; Vuckovic et al., 2020; Zhang et al., 2020; Edelman et al., 2022; Snell et al., 2021; Wei et al., 2021; Deng et al., 2023d; 2024a). Among these, Zhao et al. (2023) revealed the adeptness of mid-scale masked language models in identifying syntactic elements, paving the way for innovations like partial parse tree reconstructions. Inspired the exponential mechanism in attention structure, Gao et al. (2023) provide an analysis which shows exponential regression within the over-parameterized neural tangent kernel framework can converge. In the over-constrained setting, several work show the convergence for attention inspired regression problem (Li et al., 2023; Deng et al., 2023b).

## B $\ell_2$ SAMPLER

We give the full details of the standard $L_2$ sampler from Jowhari et al. (2011); Mahabadi et al. (2020) in Algorithm 2. The proof of correctness is verbatim from Jowhari et al. (2011); Mahabadi et al. (2020). The challenge is how to implement the data structures of $y$, which is implicitly defined as $(A_1 \otimes A_2)x$. By comparison, in the standard setting of $\ell_2$ samplers Monemizadeh & Woodruff (2010); Andoni et al. (2011); Jowhari et al. (2011); Jayaram & Woodruff (2021); Mahabadi et al. (2020), $y$ is given as a data stream.

## C FROM TAIL TO SAMPLING

**Lemma C.1** (Restatement of Lemma 7.4). *Let* $y = (A_1 \otimes A_2)x \in \mathbb{R}^{n^2}$. *Let only one of* $A_1$ *and* $A_2$ *be updated in streaming. Let* $w = \frac{y_i}{\sqrt{u_i}}$ *for a constant* $u_i \in [0,1]$ *generated uniformly at random. There is an algorithm* $\mathcal{A}$ *that that uses* $O(nd) + \mathrm{poly}\left(\frac{1}{\epsilon}, \log n\right)$ *space, uses* $O(n)$ *update time, and estimates each element of* $w$ *up to additive error* $\epsilon \cdot \|z\|_2$, *where* $z$ *denotes the tail vector of* $w$ *without the largest* $\frac{1}{\epsilon^2}$ *entries in magnitude. Specifically, for all* $i \in [n^2]$, *we have* $|\widehat{w}_i - w_i| \leq \epsilon \cdot \|z\|_2$.

---

**Algorithm 2** Standard $\ell_2$ Sampler, e.g., extension of Jowhari et al. (2011) to $p = 2$

---

1: For each $i \in [n]$, let $u_i \in [0, 1]$ be chosen uniformly at random
2: $w_i \leftarrow \frac{y_i}{\sqrt{u_i}}$
3: Let $z$ denote the tail vector of $w$ without the largest $\frac{1}{\epsilon^2}$ entries in magnitude
4: Let $\widehat{Y}$ be a 2-approximation of $\|y\|_2$
5: Let $\widehat{Z}$ be a 2-approximation of $\|z\|_2$
6: $i \leftarrow \text{argmax}_{i \in [n]} |\widehat{w_i}|$
7: Let $C > 0$ be a large constant determined by the additive faliure probability $\frac{1}{\text{poly}(n)}$
8: **if** $\widehat{Z} > \sqrt{\frac{C \log n}{\epsilon}} \cdot \widehat{Y}$ or $|w_i| < \sqrt{\frac{C \log n}{\epsilon}} \cdot \widehat{Y}$ **then**
9:    Return FAIL
10: **else**
11:    Return $i$ with estimate $\sqrt{u_i} \cdot \widehat{w_i}$
12: **end if**

---

*Proof.* Consider hash function $h_1, h_2 : [n] \to [b]$. Consider random sign functions $\sigma_1, \sigma_2 : [n] \to \{-1, +1\}$. We consider a fixed index $i_1, i_2 \in [n]$. Let $j = h_1(i_1) + h_2(i_2) \pmod{b}$. Let $h^{-1}(j)$ denote the all the pairs $(i_1, i_2) \in [n] \times [n]$ such that $h_1(i_1) + h_2(i_2) \pmod{b} = j$. Note that $\widehat{y_i}$ induced by $h$ is $\widehat{w_i} = w_i + \sum_{l \in h^{-1}(j) \setminus \{i\}} s_i s_l w_{l_1} w_{l_2}$. For ease of presentation, we write $\sigma_i = \sigma_{1,i_1} \sigma_{2,i_2}$ and $\sigma_l = \sigma_{1,l_1} \sigma_{2,l_2}$.

$$\mathbb{E}[\widehat{w_i}] = \mathbb{E}\left[w_i + \sum_{l \in h^{-1}(j) \setminus \{i\}} \sigma(i)\sigma(l)w_l\right] = \mathbb{E}[w_i] + \sum_{l \in h^{-1}(j) \setminus \{i\}} \mathbb{E}[\sigma(i) \cdot \sigma(l)] \cdot w_l$$

$$= w_i + \sum_{l \in h^{-1}(j) \setminus \{i\}} \mathbb{E}[\sigma(i)] \cdot \mathbb{E}[\sigma(l)] \cdot w_l = w_i,$$

where the first step follows from definition, the second step follows from linearity of expectation, the third step follows from $\sigma(i)$ and $\sigma(l)$ are independent, the forth step follows from $\mathbb{E}[\sigma(l)] = 0$.

We now upper bound the variance of $\widehat{w_i} - y_i$ by analyzing $\mathbb{E}[(\widehat{y_i})^2]$. Let $\mathcal{H}$ be the set of the top $\frac{1}{\epsilon^2}$ items and let $\mathcal{E}$ be the event that none of the items in $\mathcal{H}$ are mapped to $h(i)$, i.e., $h(a) \neq h(i)$ for all $a \in \mathcal{H}$.

Observe that for $b = \frac{100}{\epsilon^2}$, we have that $\Pr[\mathcal{E}] \geq 0.9$. Then we have:

$$\mathbb{E}[(\widehat{w_i} - w_i)^2 \mid \mathcal{E}] = \mathbb{E}[(\sum_{l \in [n]^2 \setminus \mathcal{H}, l \in h^{-1}(j)} \sigma(i)\sigma(l)w_l)^2] = \mathbb{E}\left[\sum_{l \in [n]^2 \setminus \mathcal{H}, l \in h^{-1}(j)} w_l^2\right]$$

$$= \frac{1}{b} \cdot \sum_{l \in [n]^2 \setminus \mathcal{H}, l \in h^{-1}(j)} w_l^2 \leq \frac{1}{b} \cdot (w_1^2 + \ldots + w_{n^2}^2 - \sum_{l \in \mathcal{H}} w_l^2)$$

$$= 100\epsilon^2 \cdot \|z\|_2^2,$$

for $b = \frac{100}{\epsilon^2}$, since $z$ is the vector corresponding to $y$ that removes the entries in $\mathcal{H}$. By Chebyshev's inequality, we have that $\Pr[|\widehat{w_i} - w_i| \geq \epsilon \cdot \|z\|_2 \mid \mathcal{E}] \leq \frac{1}{10}$. Since $\Pr[\mathcal{E}] \geq 0.9$, then $\Pr |\widehat{w_i} - w_i| \geq \epsilon \cdot \|z\|_2 \leq 0.2$, for a fixed hash function $h$. By taking the median of $O(\log n)$ estimations corresponding to $O(\log n)$ different hash functions $h$, we have that $\Pr[|\widehat{w_i} - w_i| \geq \epsilon \cdot \|z\|_2] \leq \frac{1}{n^{10}}$. Thus by a union bound over $i \in [n] \times [n]$, we have that with probability at least $1 - \frac{1}{n^5}$, we have for all $i \in [n]$, $|\widehat{w_i} - w_i| \geq \epsilon \cdot \|z\|_2$. $\square$

**Lemma C.2** (Restatement of Lemma 7.5). *Let $y = (A_1 \otimes A_2)x \in \mathbb{R}^{n^2}$ and let $w \in \mathbb{R}^{n^2}$ so that $w_i = \frac{y_i}{\sqrt{u_i}}$ for a constant $u_i \in [0, 1]$ generated uniformly at random. Let $z$ denote the tail vector of $w$ without the largest $\frac{1}{\epsilon^2}$ entries in magnitude. Let $\widehat{Z}$ be a 2-approximation to $\|z\|_2$ and $\widehat{Y}$ be a 2-approximation to $\|y\|_2$. Then*

$$\Pr\left[\widehat{Z} > \sqrt{\frac{C \log n}{\epsilon}} \cdot \widehat{Y}\right] \leq O(\epsilon) + \frac{1}{\text{poly}(n)}.$$

*Proof.* Let $\mathcal{E}_1$ denote the event that $\widehat{Z}$ is a 2-approximation to $\|z\|_2$ and $\widehat{Y}$ is a 2-approximation to $\|y\|_2$, so that

$$\Pr[\mathcal{E}_1] \geq 1 - \frac{1}{\text{poly}(n)}.$$

Conditioned on $\mathcal{E}_1$, it suffices to bound the probability that

$$4\|z\|_2 > \sqrt{\frac{C\log n}{\epsilon}} \cdot \|y\|_2.$$

Let $j \in [n^2]$ be a fixed index and let $u_j$ be fixed.

Let $T = \sqrt{\epsilon} \cdot \|y\|_2$ and for each $i \in [n^2]$, we define the indicator random variable $W_i = 1$ if $|w_i| > T$ and $W_i = 0$ otherwise, if $|w_i| \leq T$. Note that $W_i$ is an indicator random variable for whether the coordinate $w_i$ in the vector $w$ is "heavy" in magnitude.

We then define

$$Z_i = \frac{w_i^2}{T^2} \cdot (1 - W_i)$$

to be the scaled contribution of the small entries of $z$, and observe that $Z_i \in [0, 1]$.

Let

$$W = \sum_{i \in [n^2], i \neq j} w_i$$

denote the total number of heavy indices besides possibly index $j$ and $Z = \sum_{i \in [n^2], i \neq j} Z_i$ denote the total scaled contribution of the light indices besides possibly index $j$. Let $v$ denote the vector containing the heavy indices, so that $v_i = w_i$ for $W_i = 1$ and $v_i = 0$ otherwise for $W_i = 0$. Note that $v$ has sparsity at most $Y + 1$ and moreover $U^2 Z = \|w - v\|_2^2$. We also have that $\|z\|_2 \leq \|w - v\|_2$ unless $W \geq \frac{2}{\epsilon^2}$.

Let $\mathcal{E}_2$ denote the event that $W \geq \frac{2}{\epsilon^2}$ and let $\mathcal{E}_3$ denote the event that $Z \geq \frac{C\log n}{16T^2\epsilon} \cdot \|y\|_2^2$. Observe that if neither $\mathcal{E}_2$ nor $\mathcal{E}_3$ occur, then we have $4\|z\|_2 \leq \sqrt{\frac{C\log n}{\epsilon}} \cdot \|y\|_2$, as desired. Thus it remains to bound the probability of the failure events $\mathcal{E}_2$ and $\mathcal{E}_3$.

We have $\mathbb{E}[W_i] = \frac{\|w\|_2^2}{T^2}$, so that $\mathbb{E}[W] \leq \frac{1}{\epsilon}$. By Markov's inequality, we have that $\Pr[\mathcal{E}_2] \leq \frac{\epsilon}{2}$.

We now upper bound $\Pr[\mathcal{E}_3]$. Recall that $Z_i = \frac{w_i^2}{T^2} \cdot (1 - W_i) = \frac{w_i^2}{Tu_i^2} \cdot (1 - W_i)$, since $w_i = \frac{y_i}{\sqrt{u_i}}$. Observe that $Z_i > 0$ only if $|w_i| < T$, i.e., if $u_i \geq \frac{y_i^2}{\epsilon \cdot \|y\|_2^2}$, since $T = \sqrt{\epsilon} \cdot \|y\|_2$. For $\epsilon \in (0, 1)$, we thus have

$$\mathbb{E}[Z_i] \leq \int_{y_i^2/\|y\|_2^2}^{1} z_i \, du_i$$

$$= \int_{y_i^2/\|y\|_2^2}^{1} \frac{y_i^2}{u_i} \frac{1}{T^2} \, du_i.$$

Now, let $\mathcal{E}_4$ be the event that $u_i \geq \frac{1}{n^{C/2}}$ for all $i \in [n^2]$, so that $\Pr[\mathcal{E}_4] \geq 1 - \frac{1}{n^{C/2-2}}$.

Then

$$\mathbb{E}[Z_i \mid \mathcal{E}_4] \leq \frac{1}{1 - \frac{1}{n^{C/2-2}}} \int_{1/n^{C/2}}^{1} \frac{y_i^2}{u_i} \frac{1}{T^2} \, du_i$$

$$\leq \frac{C\log n}{T^2} y_i^2.$$

Thus, we have

$$
\begin{aligned}
\mathbb{E}[Z \mid \mathcal{E}_4] &= \sum_{i \in [n^2]} \mathbb{E}[Z_i \mid \mathcal{E}_4] \\
&= \sum_{i \in [n^2]} \frac{C \log n}{T^2} y_i^2 \\
&\leq \sum_{i \in [n^2]} \frac{C \log n}{\epsilon} \frac{y_i^2}{\|y\|_2^2} \\
&= \frac{C \log n}{\epsilon}.
\end{aligned}
$$

Thus by Markov's inequality, the probability that $Z$ is larger than $\frac{C \log n}{16 T^2 \epsilon} \cdot \|y\|_2^2 = \frac{C \log n}{16 \epsilon^2}$ is at most $\frac{\epsilon}{16}$. The claim then follows from taking a union bound over the events $\mathcal{E}_1, \neg \mathcal{E}_2, \neg \mathcal{E}_3, \neg \mathcal{E}_4$. $\square$

## D  TENSOR SAMPLING

**Theorem D.1** (Restatement of Theorem 7.6). *Let $y = (A_1 \otimes A_2)x \in \mathbb{R}^{n^2}$ and let $w \in \mathbb{R}^{n^2}$ so that for each $i \in [n^2]$, $w_i = \frac{y_i}{\sqrt{u_i}}$ for a constant $u_i \in [0,1]$ generated uniformly at random. Let $z$ denote the tail vector of $w$ without the largest $\frac{1}{\epsilon^2}$ entries in magnitude. Suppose there exists:*

1. *An algorithm $\mathcal{A}_1$ that provides a 2-approximation to $\|y\|_2$ with probability $1 - \frac{1}{n^2}$.*

2. *An algorithm $\mathcal{A}_2$ that provides a 2-approximation to $\|z\|_2$ with probability $1 - \frac{1}{n^2}$.*

3. *An algorithm $\mathcal{A}_3$ that estimates each element of $w$ up to additive error $\epsilon \cdot \|z\|_2$,*
$$
|\widehat{w_i} - w_i| \leq \epsilon \cdot \|z\|_2,
$$
   *for all $i \in [n^2]$.*

*Then there exists a data structure that uses $\mathrm{poly}\left(\frac{1}{\epsilon}, \log n\right)$ bits of space and outputs each index $i$ with probability $p_i$, such that*

$$
(1 - \epsilon) \cdot \frac{y_i^2}{\|y\|_2^2} - \frac{1}{\mathrm{poly}(n)} \leq p_i \leq (1 + \epsilon) \cdot \frac{y_i^2}{\|y\|_2^2} + \frac{1}{\mathrm{poly}(n)}.
$$

*Proof.* Let $i$ be fixed and let $\mathcal{E}$ denote the event that $u_i < \frac{\epsilon}{C \log n} \frac{y_i^2}{\widehat{Y}^2}$, so that $|w_i| > \sqrt{\frac{C \log n}{\epsilon}} \cdot \widehat{Y}$.

Let $\mathcal{E}_1$ denote the event that $\widehat{Y}$ is a 2-approximation to $\|y\|_2$, $\widehat{Z}$ is a 2-approximation to $\|z\|_2$, and $|\widehat{w_i} - w_i| \leq \epsilon \cdot \|z\|_2$ for all $i \in [n]$. Let $\mathcal{E}_2$ denote the event that $\widehat{Z} > \sqrt{\frac{C \log n}{\epsilon}} \cdot \widehat{Y}$ and let $\mathcal{E}_3$ denote the event that multiple indices $j$ satisfy $|w_j| > \sqrt{\frac{C \log n}{\epsilon}} \cdot \widehat{Y}$. Finally, let $\mathcal{E}_4$ denote the event that $|\widehat{w_i}| < \sqrt{\frac{C \log n}{\epsilon}} \cdot \widehat{Y}$.

Intuitively, $\mathcal{E}_1$ is a good event, i.e., correctness of the data structures, which we would like to hold. On the other hand, $\mathcal{E}_2, \mathcal{E}_3, \mathcal{E}_4$ are bad events that distort the sampling probabilities, which we would like to avoid.

We first note that $\mathcal{E}_1$ holds with high probability due to the correctness of the CountSketch and $L_2$-norm estimation data structures. We next note that by Lemma 7.5, the probability that $\mathcal{E}_2$ occurs is $O(\epsilon)$.

Next, note that the probability that for a fixed $j \in [n]$, $u_j$ satisfies $\frac{y_j^2}{u_j} \geq \frac{C \log n}{\epsilon} \cdot \widehat{Y}$ is at most $\frac{\epsilon}{C' \log n} \frac{y_j^2}{\|y\|_2^2}$ for some constant $C'$. Thus summing over all $j \in [n]$, the probability that there exist an additional $j \in [n]$ for which $|w_j| > \sqrt{\frac{C \log n}{\epsilon}} \cdot \widehat{Y}$ is $O(\epsilon)$. Thus the probability that $\mathcal{E}_3$ occurs is $O(\epsilon)$.

Finally, conditioned on $\neg \mathcal{E}_2$, we have that $\widehat{Z} \le \sqrt{\frac{C \log n}{\epsilon}} \cdot \widehat{Y}$. Then conditioning on $\mathcal{E}_1$, we have $\|z\|_2 \le \widehat{Z}$ and thus $|\widehat{w_i} - w_i| \le \epsilon \widehat{Z} \le \sqrt{C \epsilon \log n} \widehat{Y}$, so that $\mathcal{E}_4$ can only occur for $\sqrt{\frac{C \log n}{\epsilon}} \cdot \widehat{Y} \le |w_i| \le \sqrt{\frac{C \log n}{\epsilon}} \cdot \widehat{Y}$, which is at most probability $O\left(\frac{\epsilon^2}{C \log n} \frac{y_i^2}{\widehat{Y}^2}\right)$, over the randomness of $u_i$.

In summary, we observe that conditioned on some value being output, the probability that item $i$ is selected is proportional to the event that the events $\mathcal{E}$ and $\mathcal{E}_1$ occur, and none of the events $\mathcal{E}_2, \mathcal{E}_3, \mathcal{E}_4$ occur. The probability that $\mathcal{E}$ occurs is $\frac{\epsilon}{C \log n} \frac{y_i^2}{\widehat{Y}^2}$, which $u_i$ is chosen uniformly at random. Due to the event $\mathcal{E}_1$, the sampling probability is distorted additively by $\frac{1}{\text{poly}(n)}$, while due to the events $\mathcal{E}_2, \mathcal{E}_3, \mathcal{E}_4$, the sampling probability is distorted multiplicatively by $(1 + \epsilon)$. Thus conditioned on the event that some index is returned, the probability $p_i$ that index $i$ is returned satisfies

$$(1 - \epsilon) \cdot \frac{y_i^2}{\|y\|_2^2} - \frac{1}{\text{poly}(n)} \le p_i \le (1 + \epsilon) \cdot \frac{y_i^2}{\|y\|_2^2} + \frac{1}{\text{poly}(n)},$$

as desired. □

# E  LLM USAGE DISCLOSURE

LLMs were used only to polish language, such as grammar and wording. These models did not contribute to idea creation or writing, and the authors take full responsibility for this paper's content.

