# OpenReview forum: "Towards Sampling Data Structures for Tensor Products in Turnstile Streams"
_ICLR.cc/2026/Conference — ICLR 2026 Poster_

### Official Review · Reviewer_dsvo · 2025-10-28

**Soundness:** 3
**Presentation:** 2
**Contribution:** 2
**Rating:** 4
**Confidence:** 3

**Summary:**

The paper proposes attention sampling - sampling a token index from a token stream from a distribution constructed according to the attention function. It gives a space lower bound for this problem and also provides upper bounds for similar L2 sampling questions.

**Strengths:**

1. The problem considered by the paper is interesting: sampling from a distribution according to the attention mechanism. Understanding the complexity of this problem and designing efficient algorithms for can have an important impact on efficient AI algorithms.
2. The methods proposed by the paper, though they generalize prior approaches and techniques, are applied with clarity and in a set of well-motivated theoretical settings.
3. The paper is rigorous and methodical in its approach. Its algorithms are well stated and proven.
4. The paper provides an extensive literature review.

**Weaknesses:**

1. The motivation behind "attention sampling" is not very well supported. The authors state that solving this problem would be positive towards developing efficient transformers, but this is not justified via experiments or suggestions of ways their algorithms can be used in practice. And without those, it is hard to see why attention sampling would be an important problem to solve. Furthermore, the authors show an $\Omega(n)$ lower bound for exponential sampling, meaning that even if attention sampling is very useful, the best thing we can do is calculate the underlying distribution explicitly.
    * As a smaller comment, streaming space lower bounds for attention mechanisms have been proposed and analyzed before (also via communication complexity reductions). It would be nice for the paper to cite these works as well: [1], [2].
2. The L2 sampling contributions, while interesting, deviate from the main theme of the paper. Attention sampling, as proposed, should intuitively be related to the Transformer architecture or attention mechanism. Perhaps for different normalization functions (other than softmax), sampling in sublinear space is possible, which would give further motivation towards studying and using those functions. The L2 results seem like a natural extension of the well-studied problem of $L_p$ sampling in a turnstile stream, and though they make use of non-trivial sketch constructions to deal with updates in the $A$ matrix, they don't fit in very well to the paper's narrative in my opinion.

**References**
* [1] Han, I., Kapralov, M., Kochetkova, E., Sheth, K., & Zandieh, A. (2025). Streaming Attention Approximation via Discrepancy Theory. arXiv preprint arXiv:2502.07861.
* [2] Haris, Themistoklis, and Krzysztof Onak. "Compression barriers for autoregressive transformers." arXiv preprint arXiv:2502.15955 (2025).

**Questions:**

1. How are the entries of matrix $A$ updated? Is it row-by-row, or any kind of update is acceptable?

---

> ### Author Response · Authors · 2025-12-03
>
> > The motivation behind "attention sampling" is not very well supported. The authors state that solving this problem would be positive towards developing efficient transformers, but this is not justified via experiments or suggestions of ways their algorithms can be used in practice. And without those, it is hard to see why attention sampling would be an important problem to solve. Furthermore, the authors show an lower bound for exponential sampling, meaning that even if attention sampling is very useful, the best thing we can do is calculate the underlying distribution explicitly.
>
> Computing attention exactly requires $O(n^2)$ space and time in the number of tokens $n$, which limits the implementation of long-context Transformers, inspiring our work to reduce this computational overhead.
> On the other hand, attention is **empirically and theoretically sparse** in many real-world settings (e.g., Child et al., 2019; Deng et al., 2024b cited in our paper), and only a small fraction of coordinates significantly influence the output.
> Thus, our samplers directly recover the heavy coordinates in sparse, large-scale models and can be used to construct attention masks efficiently in practical scenarios.
> In addition, our sampler is suitable for turnstile streams that incorporate both positive updates and negative updates, which enables dynamic training and inference (see our response to Reviewer odpP).
>
> Regarding the exponential case, our
> $\Omega(n)$ lower bound shows that softmax sampling is inherently hard in streaming settings.
> This motivates our focus on polynomial attention, where efficient sampling is possible. Prior work such as PolySketchFormer (Kacham et al., 2023) already demonstrates that polynomial attention can match softmax empirically, making it a meaningful and useful target.
> We thank the reviewer for pointing out the concerns on the motivation and will emphasize more in our revised version.
>
> > As a smaller comment, streaming space lower bounds for attention mechanisms have been proposed and analyzed before (also via communication complexity reductions). It would be nice for the paper to cite these works as well: [1], [2].
>
> We thank the reviewer for pointing out these references.
> We will cite [1] Han et al. (2025) and [2] Haris & Onak (2025) and discuss how our attention samplers align with their results.
>
> > The L2 sampling contributions, while interesting, deviate from the main theme of the paper. Attention sampling, as proposed, should intuitively be related to the Transformer architecture or attention mechanism. Perhaps for different normalization functions (other than softmax), sampling in sublinear space is possible, which would give further motivation towards studying and using those functions. The L2 results seem like a natural extension of the well-studied problem of sampling in a turnstile stream, and though they make use of non-trivial sketch constructions to deal with updates in the matrix, they don't fit in very well to the paper's narrative in my opinion.
>
> We believe the $L_2$ sampling results fit naturally into the paper’s narrative.
> As discussed in the above responses, softmax attention is provably hard to sample in sublinear space, and recent lower bounds (e.g., Alman & Song, 2023) showed the difficulty of approximating softmax with sketches up to small entry-wise error.
> This motivates studying alternative normalization functions for which efficient sampling is possible.
>
> In this context, $L_2$ sampling corresponds to the case of polynomial attention, a family that has gained traction as a practical softmax substitute (e.g., PolySketchFormer).
> Our contribution goes beyond classical $L_2$ sampling by handling streaming updates to both the input matrix $A$ and the weight matrix $x$, which aligns exactly with the matrix-vector multiplication in the attention mechanism, and we also consider a tensor version of the problem.
>
> > How are the entries of matrix $A$ updated? Is it row-by-row, or any kind of update is acceptable?
>
> This depends on the setting.
> In Section 5, we consider sampling the heavy coordinates of $Ax$, where $A$ and $x$ are updating entry-wise (see e.g., lines 340-341).
> In Section 7, we consider a practical scenario in inference, where $A_1$, $A_2$ are the input matrices and they are updating row-by-row, and $x$ is the fixed, precomputed weight vector.

---

### Official Review · Reviewer_odpP · 2025-10-30

**Soundness:** 3
**Presentation:** 2
**Contribution:** 2
**Rating:** 6
**Confidence:** 2

**Summary:**

The paper studies and characterizes a polynomial type sampling algorithms for the application of streaming LLMs where the model is
expected to engage in large inference sequence generations (as a long chat bot conversation).
Moreover, the authors derive a lower bound for the softmax distribution arguing for its "hardness", as well as upper and
lower bounds for polynomial type samples.

**Strengths:**

* The paper contains a comprehensive theoretical analysis for polynomial type samples proving several bounds and some
  lower bounds.

**Weaknesses:**

* The paper is motivated by the computational challenges of large-scale attention-based models which I'm familiar with.
However, the paper does not motivate why are we using particularly turnstile streams.
Some clearer motivation of a general Machine Learning audience would be highly valuable.
Moreover, Definition 1.1 talks about the attention sample for a matrix that is not square on the sequence length, which
deviates from the usual attention definition.
* I understand that the paper is a theoretical one where a thorough analysis is done to some polynomial type of
  algorithms. However, no empirical demonstration is done to corroborate its the core question of the paper:
  "instead of computing all the entries, can we recover the most important ones in efficient space and time?"
* I'm willing to revisit my score if the motivation for the work and its applicability is clearer to me.

**Questions:**

* In the abstract you claim: "Our approach significantly reduces the computational burden of traditional attention mechanisms"
Could you elaborate why this is true. My understanding is that the scope of streaming LLMs is not commonplace and
therefore the claim appears to encompass a wider scope. Am I missing something?
* In the conclusion you claim: "our framework identify the critical components in attention computation" could you
  succinctly state what does critical components are?
* Could any of this work be applied to pre-training a model or it can only be applied to inference?
* I might've missed this motivation, but what is the value of having a tensor version of the problem? What do we gain?
* Line 049. What do you mean by first asked? "A well-known example is the \ell_2 sampler first asked by" Maybe typo.

---

> ### Author Response · Authors · 2025-12-03
>
> > The paper is motivated by the computational challenges of large-scale attention-based models which I'm familiar with. However, the paper does not motivate why are we using particularly turnstile streams. Some clearer motivation of a general Machine Learning audience would be highly valuable. Moreover, Definition 1.1 talks about the attention sample for a matrix that is not square on the sequence length, which deviates from the usual attention definition.
>
> Due to the memory constraint in computing large-scale, dynamic attention-based models, one cannot store the entire model and recompute it at each update.
> This motivates the streaming LLMs (Xiao et al., 2024), where sequences grow indefinitely, and KV caches must handle updates efficiently without recomputing everything.
> Therefore, we consider the attention sampler in the streaming setting.
>
> Additionally, turnstile streams naturally capture settings where data must be updated, not only appended.
> A key example is **machine unlearning**, where we need to remove or negate the contribution of specific data points without retraining the entire model, which aligns exactly with the turnstile model.
> In the turnstile model, attention matrices need to incorporate both positive updates (new tokens or training examples) and negative updates (removing or editing previous tokens, retracting user-provided data, or forgetting specific training segments). Turnstile streams provide the right abstraction for reasoning about these operations efficiently.
> We will add this motivation in the revised version.
>
> Regarding Definition 1.1, the non-square $A \in \mathbb{R}^{n \times d}$ reflects the tensor product view of attention ($A = A_1 \otimes A_2$, where $A_1$ and $A_2$ are the input matrices), and the vector $x$ is the vectorized version of the fused key and query matrix.
> Thus this matrix-vector multiplication $Ax$ is equivalent to the usual attention definition.
> We explained this tensor product construction in line 59-66 below Definition 1.1.
>
> > I understand that the paper is a theoretical one where a thorough analysis is done to some polynomial type of algorithms. However, no empirical demonstration is done to corroborate its the core question of the paper: "instead of computing all the entries, can we recover the most important ones in efficient space and time?"
>
> Yes, we agree that empirical work would strengthen the applications of the paper. We can demonstrate simple experiments showing that we can recover important entries via $L_2$ sampling in polylogarithmic space, in contrast to softmax's $\Omega(n)$ lower bound. We will incorporate these experiments into future version of the paper. However, we believe that showing the effects of these important entries in large-scale downstream applications is perhaps beyond the scope of this paper.
>
> > In the abstract you claim: "Our approach significantly reduces the computational burden of traditional attention mechanisms" Could you elaborate why this is true. My understanding is that the scope of streaming LLMs is not commonplace and therefore the claim appears to encompass a wider scope. Am I missing something?
>
> Thanks for the question. Due to inherent barriers (such as those proven in our paper), there has not been much progress in streaming LLMs. However, a recent work by Han et. al. [HKK+25] appearing as a spotlight presentation at NeurIPS 2025 showed that under assumptions that each row norm is bounded in the query, value, and key matrices, it is possible to achieve non-trivial guarantees using discrepancy theory. Thus our conceptual message should be considered from a similar perspective. Namely, existing approaches all require storing the entire embeddings, whereas our approach shows that the space can be significantly compressed.
>
> [HKK+25] Insu Han, Michael Kapralov, Ekaterina Kochetkova, Kshiteej Sheth, Amir Zandieh: BalanceKV: KV Cache Compression through Discrepancy Theory. NeurIPS (2025)

---

> ### Author Response · Authors · 2025-12-03
>
> > In the conclusion you claim: "our framework identify the critical components in attention computation" could you succinctly state what does critical components are?
>
> The critical components are the heavy-hitters in the attention computation $Ax$, namely, the entries in the matrix with large magnitude that capture most of the weight of the matrix-vector product. Our sampler identifies these entries without computing the full matrix.
>
> > Could any of this work be applied to pre-training a model or it can only be applied to inference?
>
> Our results apply to both pre-training and inference. The key reason is that our model works at the level of general matrix–vector computations that underlie attention mechanisms. In pre-training, for dynamic datasets (e.g., online data streams), it can sample important attention coordinates during gradient updates. In inference, it is suitable for streaming/long-context generation by maintaining samplers over evolving KV caches.
>
> > I might've missed this motivation, but what is the value of having a tensor version of the problem? What do we gain?
>
> As we mentioned in the first question, the model in our Definition 1.1 focuses on sampling the heavy coordinates in $Ax$, where $A = A_1 \otimes A_2$ is the tensor product.
> In Section 5, we consider entry-wise update of $A$ and $x$.
> As mentioned in lines 410-416, the tensor version in Section 7 considers a practical scenario in real-time inference, where the weight matrix $x$ is precomputed and fixed, and we are updating the input matrices $A_1$ or $A_2$ row-by-row.
> That is, the rows of matrix $A_1$ arrive as a data stream, representing real-time data queries.
>
> > Line 049. What do you mean by first asked? "A well-known example is the \ell_2 sampler first asked by" Maybe typo.
>
> This is a typo; it should be "first posed as a question by" (referring to Cormode et al., 2005). We will correct it in revision.

---

### Official Review · Reviewer_XQKc · 2025-10-31

**Soundness:** 2
**Presentation:** 2
**Contribution:** 2
**Rating:** 4
**Confidence:** 2

**Summary:**

The authors propose to use attention sampler for efficient attention computation in the streaming setting. They primarily provide theoretical time & space lower and upper bound for L2 sampler. The authors also extend the analysis to the case of tensor multiplication.

**Strengths:**

The theoretical investigation seems solid and also quite comprehensive. They also extend the analysis to tensors, which might shed light on some tensor attention applications.

**Weaknesses:**

- Line 086-090: bullet point 1 and 2 are somewhat overlapping and provide different claims on complexity for the same underlying setting on A and x.
- The primary problem is that there are no experiments to demonstrate actual efficiency improvement in a Transformer, even though the paper is clearly motivated by solving the efficiency challenge of large-scale attention-based models computation.

**Questions:**

N/A

---

> ### Author Response · Authors · 2025-12-03
>
> Reviewer XQKc
> > Line 086-090: bullet point 1 and 2 are somewhat overlapping and provide different claims on complexity for the same underlying setting on A and x.
>
> Thank you for pointing this out.
> The second bullet point should be "for updating $x$ and fixed $A$".
> We will correct it in the revised version.
>
> > The primary problem is that there are no experiments to demonstrate actual efficiency improvement in a Transformer, even though the paper is clearly motivated by solving the efficiency challenge of large-scale attention-based models computation.
>
> We appreciate the reviewer's positive evaluation on our motivation.
> Our paper is primarily theoretical, focusing on proving space and time guarantees for our proposed attention samplers in turnstile streams.
> Implementing these samplers in full Transformer pipelines (e.g., as sparse masks or KV-cache compressors) requires substantial system integration and training, which we view as valuable future work rather than the scope of this paper.
>
> On the other hand, we want to emphasize the practical usage of our attention sampler.
> Prior works such as PolySketchFormer (Kacham et al., 2023) demonstrate the empirical promise of polynomial attention, and our results provide the theoretical foundations for sampling polynomial attention.
> For sparse attention mechanisms, our sampler can efficiently recover heavy hitters, which can then be used as the active attention set in subsequent computation.
> For streaming attention, our sampler maintains a compact summary of past tokens and can produce attention samples at any time point, which provides efficient method for real-time monitoring.
> We will strengthen the discussion in the revised version on these aspects.

---

### Meta-Review · Area_Chair_fVLa · 2025-12-20

**Summary:**

This paper provides a rigorous theoretical analysis of attention sampling in turnstile streams, offering valuable space lower bounds for softmax attention and efficient upper bounds for polynomial type samplers. The reviewers consistently recognized the solidity of the theoretical proofs. While the lack of empirical experiments was raised as a concern, the paper's contribution lies in establishing fundamental limits and algorithmic possibilities. The authors also effectively clarified the motivation for the turnstile setting (e.g., machine unlearning) during the rebuttal. Therefore, I recommend acceptance.

By the way, I think the reference: James Manyika. An overview of bard: an early experiment with generative ai. is wrong. Based on google scholar, it should be: Manyika, James, and Sissie Hsiao. "An overview of Bard: an early experiment with generative AI."
You missed an author, please fix this citation.

**Reviewer Concerns:**

Addressed:
1. Typo in Complexity Claims in lines 086-090.
2. Motivation for Turnstile Streams. The authors clarified that turnstile streams are essential for modeling machine unlearning and dynamic KV-cache updates in streaming LLMs without full re-computation.
3. Non-Square Matrix Definition. The authors explained that the non-square matrix A in Definition 1.1 is consistent with the tensor product view and vectorization used in the attention mechanism.
4. Missing Citations: The authors agreed to cite the suggested works by Han et al. and Haris & Onak in the revision.

Outstanding:
The main concern about this paper is lack of empirical validation.

**Reviewer Scores:**

1. Reviewer XQKc may not update the score because he/she provides little information in the review.
2. Reviewer odpP may increase score because the authors solved many of his/her questions in the rebuttal.
3. Reviewer dsvo may not update the score because he/she have the concern that the polynomial attention is not as useful as the softmax version.

---

### Decision · Program_Chairs · 2026-01-26

Accept (Poster)